# Evaluating the Performance of Machine Learning Approaches to Predict the Microbial Quality of Surface Waters and to Optimize the Sampling Effort

**Manel Naloufi** [1,2,*], **Françoise S. Lucas** [2], **Sami Souihi** [3], **Pierre Servais** [4], **Aurélie Janne** [5] **and Thiago Wanderley Matos De Abreu** [3,*]

1 Direction de la Propreté et de l'Eau, Service Technique de l'Eau et de l'Assainissement, 27 rue du Commandeur, 75014 Paris, France

2 Laboratoire Eau, Environnement et Systèmes Urbains (Leesu), Université Paris-Est Créteil, École des Ponts ParisTech, 61 Avenue du Général de Gaulle, CEDEX, Créteil, 94010 Paris, France; lucas@u-pec.fr

3 Image, Signal and Intelligent Systems (LiSSi) Laboratory, University of Paris-Est Créteil Val de Marne, 122 rue Paul Armangot, Vitry sur Seine, 94400 Paris, France; sami.souihi@u-pec.fr

4 Ecology of Aquatic Systems, Université Libre de Bruxelles, 50 Av. Franklin Roosevelt, 1050 Brussels, Belgium; Pierre.Servais@ulb.be

5 Syndicat Marne Vive, Maison de la Nature, 77 quai de la Pie, Saint-Maur-des-Fossés, 94100 Paris, France; aurelie.janne@marne-vive.com

\* Correspondence: manel.naloufi@paris.fr (M.N.); thiago.wanderley-matos-de-abreu@u-pec.fr (T.W.M.D.A.)

**Abstract:** Exposure to contaminated water during aquatic recreational activities can lead to gastrointestinal diseases. In order to decrease the exposure risk, the fecal indicator bacteria *Escherichia coli* is routinely monitored, which is time-consuming, labor-intensive, and costly. To assist the stakeholders in the daily management of bathing sites, models have been developed to predict the microbiological quality. However, model performances are highly dependent on the quality of the input data which are usually scarce. In our study, we proposed a conceptual framework for optimizing the selection of the most adapted model, and to enrich the training dataset. This framework was successfully applied to the prediction of *Escherichia coli* concentrations in the Marne River (Paris Area, France). We compared the performance of six machine learning (ML)-based models: K-nearest neighbors, Decision Tree, Support Vector Machines, Bagging, Random Forest, and Adaptive boosting. Based on several statistical metrics, the Random Forest model presented the best accuracy compared to the other models. However, 53.2 ± 3.5% of the predicted *E. coli* densities were inaccurately estimated according to the mean absolute percentage error (MAPE). Four parameters (temperature, conductivity, 24 h cumulative rainfall of the previous day the sampling, and the river flow) were identified as key variables to be monitored for optimization of the ML model. The set of values to be optimized will feed an alert system for monitoring the microbiological quality of the water through combined strategy of in situ manual sampling and the deployment of a network of sensors. Based on these results, we propose a guideline for ML model selection and sampling optimization.

**Keywords:** water quality prediction; machine learning; *Escherichia coli* concentration; optimized sampling; river monitoring

## 1. Introduction

Worldwide the heat wave episodes have recently intensified the development of aquatic recreational activities in megapoles, increasing the interactions between citizens and freshwater in urban context [1]. Indeed, many cities, such as Paris, London, or Berlin, promote the opening of bathing areas and organize open water swimming competitions in their rivers. However, the development of these activities increases the risk of exposure of bathers to waterborne pathogens, which could result in gastrointestinal diseases, eye infections or skin irritations [2–4].

In highly urbanized areas, the microbiological quality of surface waters is strongly degraded by different diffuse and point sources of contamination that may bring high pathogen flow into the rivers [5–8]. Fecal contaminations due to sewer discharges, animal feces, and rain runoff are among the main factors impacting the quality of surface waters [9]. As the climate change is expected to modify precipitation patterns, with higher frequency of extreme events, these new conditions should negatively impact the water quality [10]. Currently, the water quality is mainly assessed using a collection of water samples for biological and chemical analysis in the laboratory and/or highly accurate sensors at fixed position. The regulatory monitoring of the bathing waters is based on the enumeration of culturable fecal indicator bacteria, *Escherichia coli* and intestinal enterococci (e.g., European Bathing directive 2006/7/EC). Such surveys are costly, time-consuming, and labor-intensive, as a consequence weekly or monthly sampling strategies are routinely implemented with additional event-based sampling [11,12].

For the daily management of urban bathing sites, models could also be used instead of collecting additional samples to check the microbial quality of the water after each short-term pollution event [11]. However, building, training, and validation of predictive models require high accuracy data that are difficult and expensive to collect [13]. Environmental stressors such as physico-chemical, hydrological, and meteorological variables are often used as input data in models to predict the concentration of fecal indicator bacteria as real-time measurement of these parameters provides cost effective and high quality data [14–16]. Among the different predictive models, machine learning tools have been proved to predict surface water quality in rivers with high accuracy in different situations, including traditional machine learning models or ensemblist methods [17–19]. However, due to the small size of most stakeholder datasets, the performance of the model can be low (see, e.g., in [17,20]). The determination of the minimum sampling size and the appropriate sampling strategy required for building, training, and validation of models is thus crucial [11].

Several strategies to improve the input dataset of machine learning models exist, however their usefulness for rationalizing the data acquisition for water quality prediction still needs to be evaluated. First, the most relevant observations during the learning process of the models could be identified in order to maximize the information gain [21]. Second, the weakness in the training dataset could be determined in order to identify which and how much additional data are needed to improve the model performance. For instance, active learning is a method that gives flexibility to identify which instances need to be added to the training set [22]. Another popular strategy is to use uncertainty sampling, to identify the point where the prediction is uncertain in the model [23]. Third, another way to enhance the amount of training data is to deploy on site a large number of low cost sensors. Each individual sensor may present a slightly greater error margin than the costly high precision equipment, however the multitude of sensors allows to build a dense network which in average is capable of providing enough information for the machine learning models [24]. However, enrichment of training datasets with high quality data of extreme events is particularly important in the context of climate change with the expected rise of temperature and increase in the frequency and intensity of storm events [12]. Therefore, the objective of this study is to explore these three strategies to improve the input datasets for training and testing machine learning models, particularly study the relevance of the active learning strategy. The ultimate goal is to provide a conceptual framework and an operating mode to assist the stakeholders in the daily management of the bathing sites. The framework thus includes (i) a guideline for selecting from a toolbox of six machine learning models, the one most adapted to their bathing site context and (ii) a strategy to improve the training and testing of their model via the sub-optimization of the sampling strategies. The Marne River (Paris Area, France) was considered as a use case. Indeed, several municipalities wish to open bathing sites on the border of the Marne river by 2022. Environmental stressor dataset used to predict *E. coli* concentrations were acquired from the Syndicat Marne Vive.

Using this database, we tested the following strategy:

(1) We propose to compare the performance of six machine-learning models, including three traditional models and three ensemblist models, to predict the concentrations of the fecal indicator bacteria *Escherichia coli*. To train and test the models, meteorological data and river flow data should be aggregated with physico-chemical data.

(2) For the chosen model, we propose to set up an alert system on the performance of the model in order to optimize the data collection. This alert should consist in identifying under which conditions the model fails to make the prediction and thus alerting the managers to carry out on site analysis in order to enrich the database.

(3) The usefulness of a network of low-cost sensors for sampling optimization as a complementary strategy to improve the dataset is discussed.

## 2. Materials and Methods

### 2.1. Study Site and Water Quality Data Collection

From mid-June to mid-September for 5 years (2015, 2017–2020) samplings were carried out weekly or bi-weekly by the Syndicat Marne Vive (SMV) on 18 stations (SMV0 to SMV17) in the lower Marne River (France) (Figure 1). For each sampling sites the following parameters were measured: *E. coli* concentrations (Most Probable Number or MPN/100 mL), temperature (°C), turbidity (FTU), conductivity (µS/cm), Total Suspended Solids or TSS (mg/L), $NH_4^+$ (mg of N/L), Total Kjeldahl Nitrogen or TKN (mg of N/L), number of dry days after the last rainfall, 24 h cumulative rainfall of the day (mm), 24 h cumulative rainfall of the previous day (mm), and the river flow (m$^3$/s) measured at Gournay-sur-Marne (Paris area, France). The sampling protocole for surface water was carried out according to the French standard FD T 90-523-1 (2008) for physicochemical parameters and according to the 2006/7/EC directive for *E. coli* concentrations. Microbiological and physico-chemical measurements were respectively carried out by Aquamesures and Eurofins (2015) and the Val de Marne Departmental Environmental Health Laboratory (2017–2020) following the French standard methods NF EN ISO 9308-3, NF EN ISO 7027-1, NF EN 27888, NF EN 872, NF T 90-015-2, NF EN 25663.

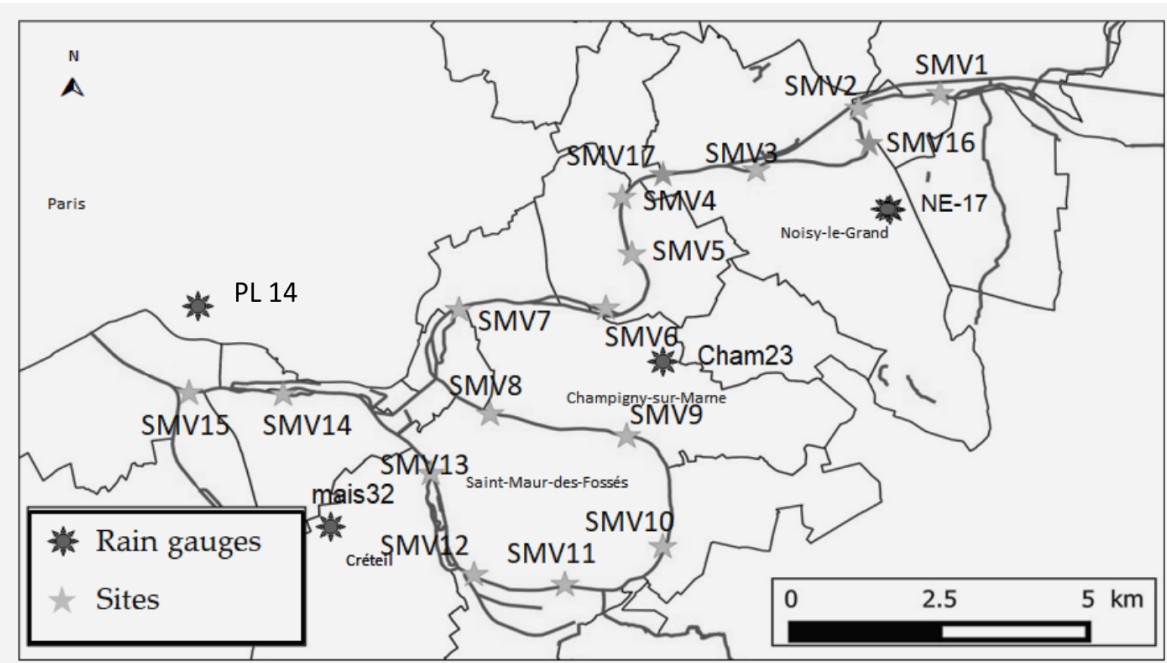

**Figure 1.** Marne River water quality monitoring stations. The light gray stars indicate the SMV sampling stations and the dark gray stars indicate the location of the rain gauges used (SMV).

Rainfall data were obtained from the network of rain gauges of the Departmental Councils of Val-de-Marne (station CHAM23, MAIS32), Seine-Saint-Denis (station NE-17),

and the City of Paris (station PL14). For each sampling point, the meteorological data of the nearest measuring station were used. For the year 2020, rainfall data of the stations SMV5 to SMV13 were not yet available. Flow data measured at the Gournay-sur-Marne station were retrieved from the Banque Hydro (http://www.hydro.eaufrance.fr/, accessed on 28 April 2021).

### 2.2. Data Preparation

A total of 1696 measures were obtained after data cleaning which consisted of removing the entries with missing and aberrant values. The ID of the station (ordered from upstream to downstream) and the ten measured physico-chemical and hydro-meteorological parameters were used as inputs for our modeling. The output of the models was the concentration of *E. coli*. Then, the dataset was divided randomly in two parts: the training set (90%, 1526 observations) and the test set (10%, 170 observations).

In order to keep all the input parameters with the same degree of influence on the final outcomes, we performed a Z-score standardization for each feature of the datasets (inputs and output) [20]. The training dataset was used for the standardization in order to block access to the values of the test set during the training of the models.

### 2.3. Machine-Learning Models

In order to evaluate the performance of the estimation of *E. coli* concentration by the machine learning models, three traditional machine-learning models (KNN (K-nearest neighbors [25]), DT (Decision tree [26]), and SVM (Support vector machines [27])) and three ensemblist learning models (bagging [28], RF (Random forest [29]), and AdaBoost (adaptive boosting [30])), that combines several base models, were selected and used in this study. All the models were carried out in python 3.7.10 with the Scikit-learn packages [31]. The GridSearchCV technique was applied to select the hyperparameter that gives the most optimal model by 5-fold cross-validation, over a parameter grid. A 10-fold cross-validation was used to train and to estimate the performance of each model, by randomly generating 10 different subsets of the training and test datasets.

#### 2.3.1. KNN

The k-nearest neighbor method consists in considering the k-nearest samples in the training dataset as an input to predict each new observation [32]. For each test datum, the closeness to all the training data is calculated with an Euclidean distance. This allows finding the k observations closest in input space to assign the test datum to a class label, and the output value of each class label is used to estimate the value to predict. The value of k thus varied from 1 to 30 with a step of 2, depending on the dataset.

#### 2.3.2. SVM

The support vector machine goal is to find the optimal hyper-plan from which the distance to all the data point is minimum, it can be applied to classification and regression problems. It consists in transforming the training data representation space into a higher dimensional space, infinite in some cases, and in constructing a hyperplane or set of hyperplanes in a high dimensional space [32]. The idea is to find a solution to flatten the projections of the training points in space without moving too far away from the training points.

#### 2.3.3. DT

Tree-based models are used to estimate a quantitative variable or classify observations by reapeatedly separating data into mutually exclusive groups. The tree-based method slices the variable space and recursively partitions each variable into subsets based on the values of the input variable and then fits a model in each of them [32].

### 2.3.4. Bagging

Bagging, also known as bootstrap aggregation, uses portions of the data and combines them by generating random subsets of the data through sampling, with repositioning [33]. The prediction is obtained by averaging the outcomes of all models. The goal is to reduce the overfitting of predictions in the model.

### 2.3.5. RF

Random forests combine multiple DT at training time. Each tree uses a sample obtained by bootstrap. Given a training set with N measures, the bootstrap aggregation randomly selects N samples with replacement of the training set [20]. Then, a subset of features is randomly selected, in order to construct a collection of decision trees with controlled variance, and fits trees to these samples. The results of the predictions from each tree are averaged [32].

### 2.3.6. Adaboost

Adaboost repeatedly uses a regression tree developed sequentially on a training sample with weights for each observation adjusted as they are developed [34]. It starts with fitting a regression to the original dataset and then adjusts the weights of the variables based on the error of the prediction. Thus, subsequent regressors focus more on poorly fitted or poorly predicted observations [32]. Finally, the results from each weak machine learning model are combined using the weighted median.

### *2.4. Models Evaluation*

In order to select the model that performs the best in predicting *E. coli* concentration, the testing phase was carried out with 10 random trials for each model with the 10 test datasets. The prediction performances of each model was evaluated by four statistical metrics. They included root mean square error (RMSE) [19], mean absolute error (MAE) [15], the ratio of performance to deviation (RPD) [35], and mean absolute percentage error (MAPE) [36,37]. These metrics are calculated as follows:

$$\text{RMSE} = \sqrt{\frac{1}{N}\sum_{i=1}^{N}(y_i - y_i')^2} \tag{1}$$

$$\text{MAE} = \frac{1}{N}\sum_{i=1}^{N}|y_i - y_i'| \tag{2}$$

$$\text{RPD} = \frac{\text{SD}}{\text{RMSE}} \tag{3}$$

$$\text{SD} = \sqrt{\frac{\sum_{i=1}^{N}(y_i - \bar{y})^2}{N}} \tag{4}$$

In these formulas, $(y_i)$ is the measured value, $(y_i')$ is the predicted value, $(N)$ is the total number of samples, and (SD) is the standard deviation of the tested dataset ($\bar{y}$ is the mean of the measured values). The smaller the RMSE or the MAE, the more stable is the predictive capacity of the model. RPD values < 1.4 indicate that the model is not reliable. For RPD values between 1.4 and 2, the model is moderately accurate and when the value is higher than 2 the model presents a high level of predictive ability [35]. Mean absolute percentage error (MAPE), which measures the goodness of fit, was also applied.

$$\text{MAPE} = \frac{|y_i - y_i'|}{y_i} * 100 \tag{5}$$

The lower the MAPE value, the more accurate is the prediction [38]. Values <50% can be evaluated as "reasonable" even good if <20%. MAPE values greater than 50%, are

indicative of an "inaccurate" prediction. A MAPE value of 50% indicates an overestimation or an underestimation of 50% with regard to the measured value.

### 2.5. Identification of the Weakness Parts of the Dataset

The MAPE values calculated during the 10 trials were used to separate the predicted values in two datasets: the reasonable (MAPE < 50%) and inaccurate estimations of the *E. coli* densities (MAPE ≥ 50%), generated by the best model on the Marne River dataset. In order to determine the physico-chemical and hydro-meteorological parameters that potentially influenced the predictive capacity of the best model, a spearman-correlation analysis was performed between the physico-chemical or hydro-meteorological parameters and the predicted values of *E. coli* (V3.5.1 [39]). All Spearman coefficients ($r_s$) were tested for their significance based on 5% error. Then, the correlation coefficients obtained with the reasonably and inaccurately predicted concentrations were compared using a unpaired two sided *t*-test in order to identify the set of hydro-meteorological and physico-chemical parameters that are influent in the model (significant $r_s$) and that need improvement (*t*-test, *p*-value < 0.01), or parameters that are less influent (nonsignificant $r_s$) but could be worth checking after improvement (*t*-test *p*-value < 0.01). Next, we identified for each parameter that could be improved (*t*-test *p*-value < 0.01), which data were weakly represented in the dataset. For each parameter, the 10 test sets have been merged together. The set of values contributing to the reasonable dataset were identified and the set of values that gave at least one inaccurate prediction were removed and inspected to identify which additional data are needed to improve the model. This allowed us to identify the set of values that give at least a reasonable or good prediction for our dataset. The guideline for selecting the best model for *E. coli* concentration prediction among the six machine learning models, and the strategy to identify a set of parameters and values range needed to optimize the sampling strategies are displayed in the Figure 2. The python and R script of the framework are available on GitHub (https://github.com/naloufi-manel/ML-performance-microbial-quality.git, accessed on 15 July 2021).

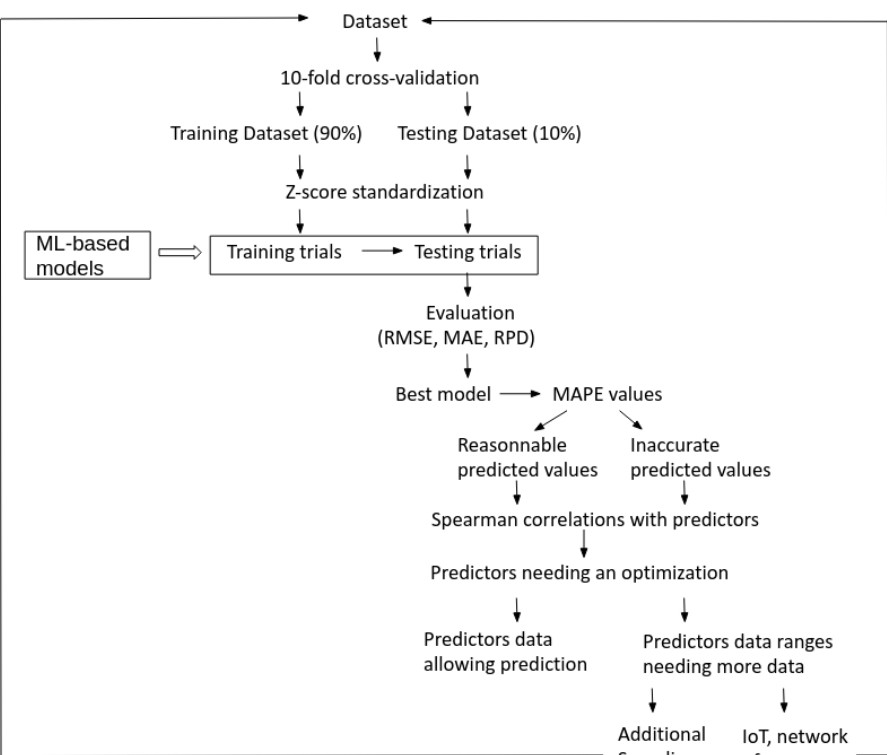

**Figure 2.** Guideline to provide and select an adapted model for water quality prediction and for the identification of a set of data to optimize the sampling strategies.

## 3. Results and Discussion

### 3.1. The Dataset Used in This Study

The Marne River dataset was characterized by a high heterogeneity concerning the number of observations per station (13 to 47 entries). The summary sample statistics of the dataset are reported in Table A1 in Appendix A. The temperature and the conductivity displayed a fair representativeness (Figure 3). However, most parameters presented a skewed distribution and the presence of upper and lower outliers (Figure 3). Indeed, for each parameter (except the temperature and the conductivity) a range of values were rarely measured and therefore not well represented in the dataset. This indicates that our dataset is not yet representative of all possible measurements.

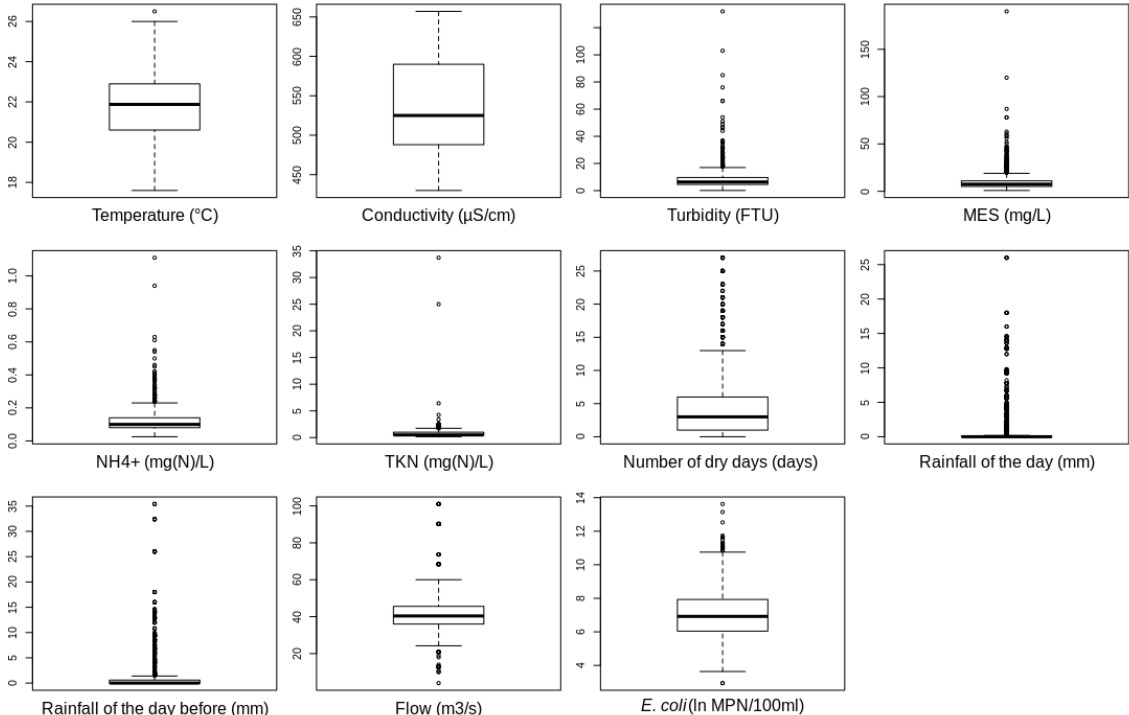

**Figure 3.** Distribution of the data for each variable. The median is indicated as a solid black line inside each boxplot, outliers are indicated as black dots. On the ordinates are the values taken by each variable with the units specified in parenthesis.

The concentration of *E. coli* (4337.61 ± 25,983.50 MPN/100 mL) measured during the 5 summers in the Marne river ranged from 19 to 820,670 MPN/100 mL. Three pollution events producing very high concentrations of *E. coli* could be identified. For instance, the maximum *E. coli* value (820,670 MPN/100 mL), corresponds to high values of turbidity, TSS and TKN levels (respectively, 28 FTU, 33 mg/L, and 2.6 mg of N/L) compared to the majority of the measurements. Extreme pollution events are often under-represented in the datasets due to their low frequency. For instance rainfalls >10 mm which lead to peaks of pollution occur less than 20 days per years in Quebec region [40]. However, removing extreme values from the dataset can lead to a decrease in the predictive capacity of the model during events with high pollution. Chen et al. [20] have shown that a better performance could be achieved after increasing the training data for each of the learning models. Considering the biased distribution of most parameters in the Marne River dataset, it may be necessary to add additional measurements to increase the size of the database and improve the training of the ML models. This would provide a better representation of the set of possible values. However, the high cost of field sampling and laboratory analyses for monitoring microbiological quality (about 100 € according to the Syndicat Marne Vive) requires an optimization of the collection in order to identify the necessary measures to efficiently complete the datasets.

### 3.2. ML-Based E. coli Prediction Comparison

Various machine learning models have been used previously to predict water quality and their predictive performance was compared to other models by assessing their ability for prediction (see, e.g., in [15,18,41]). In this study, we compared the performance of six machine learning-based algorithms (KNN, DT, SVM, Bagging, RF, and AdaBoost) to predict *E. coli* concentration in an urban river, to identify the best suited model. We performed a trial-and-error procedure, using the RMSE, MAE, and RPD metrics to evaluate the performance of each model. Average values of these statistics metrics for each random trial are available in Table A2. The RF model exhibited the highest prediction power among all the models with the weakest error (average value 0.37 ± 0.20 for RMSE and 0.09 ± 0.02 for MAE) followed by KNN and Bagging (respectively 0.41 ± 0.28 and 0.38 ± 0.19 for RMSE and 0.09 ± 0.03 and 0.14 ± 0.06 for MAE) (Figure 4).

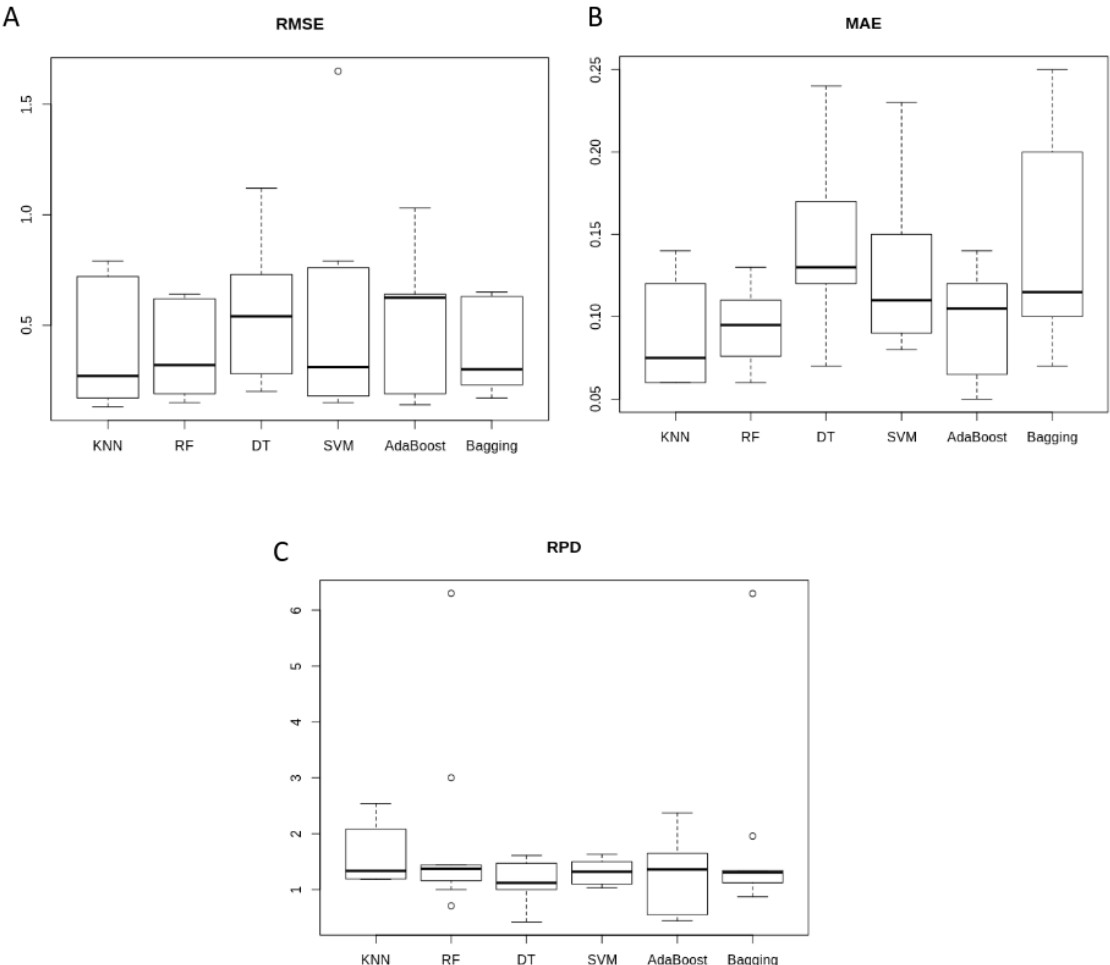

**Figure 4.** Evaluation of the prediction performances of the 6 machine learning models during the 10 trials. On the abscissa the model is indicated and on the ordinate the value of the statistical metrics are displayed (dimensionless): (**A**) RMSE, (**B**) MAE, and (**C**) RPD.

An analysis of the accuracy and reliability of the model was also performed using the RPD metrics (Figure 4C). Three models (KNN, Bagging and RF) were estimated as moderately accurate and presented acceptable results. The 3 other models were not reliable, with an RPD < 1.4 (Figure 4). For the RF model, the RDP value was close to 2 (1.91 ± 1.65), indicating that the model had a high predictive capacity. In conclusion, the RF model gave better *E. coli* concentration estimation compared to other machine learning models. This result is in agreement with Bui et al. [15] but in disagreement with the results of Chen et al. [20]. Both studies compared the performances of DT models with RF models in

their ability to predict water quality. We also checked if the performance of the models will increase by compacting the sampling sites together, however without the station ID the performance of all models slightly decreased (data not shown).

Our results confirm that Ensemblist learning models have a better performance compared to traditional models (e.g., KNN and SVM). This conclusion is in agreement with some previous studies (see, e.g., in [15,42]). However, we must keep in mind that the performance of a model depends on external uncertainty conditions [20]. Thus, for each specific dataset, several algorithms should be tested in order to find the models with the best fitting to *E. coli* concentrations. Indeed, Mälzer et al. [18] found that the performance of models could differ from one site to another along the Ruhr River in Germany. For this reason, we proposed this set of six machine learning models as a basic toolbox to be used.

### 3.3. Limits of ML-Based E. coli Estimation

Identifying observations with uncertain predictions is an approach to determine the set of data requiring optimization and thus find a way to optimize the collection and to efficiently complete our training set, allowing for a better prediction in the future by re-running the model with the newly collected measurements. Indeed, recent studies have shown that increasing the quality and quantity of the dataset by adding complementary measures allows to effectively increase the training set and to improve prediction accuracy [20,43].

To further analyze the performance, the MAPE indice, which measures the goodness of fit and examines the performance of models based on their tendency to estimate the *E. coli* concentration, was calculated for all testing trials. For 46.7 ± 3.5% of the predicted values generated by the RF model, the percentage of the absolute error was less than 50%, which indicates that the estimates were reasonable or even good. The remaining 53.2 ± 3.5% of the predicted values were associated with MAPE values equal or exceeding 50%, corresponding to inaccurate estimates. These results indicate that the RF-based model did not properly predict *E. coli* values in all contexts and that our dataset is not sufficient to efficiently train the RF-based model. Figure 5 indicates uncertainty in the prediction for some of the *E. coli* measurements.

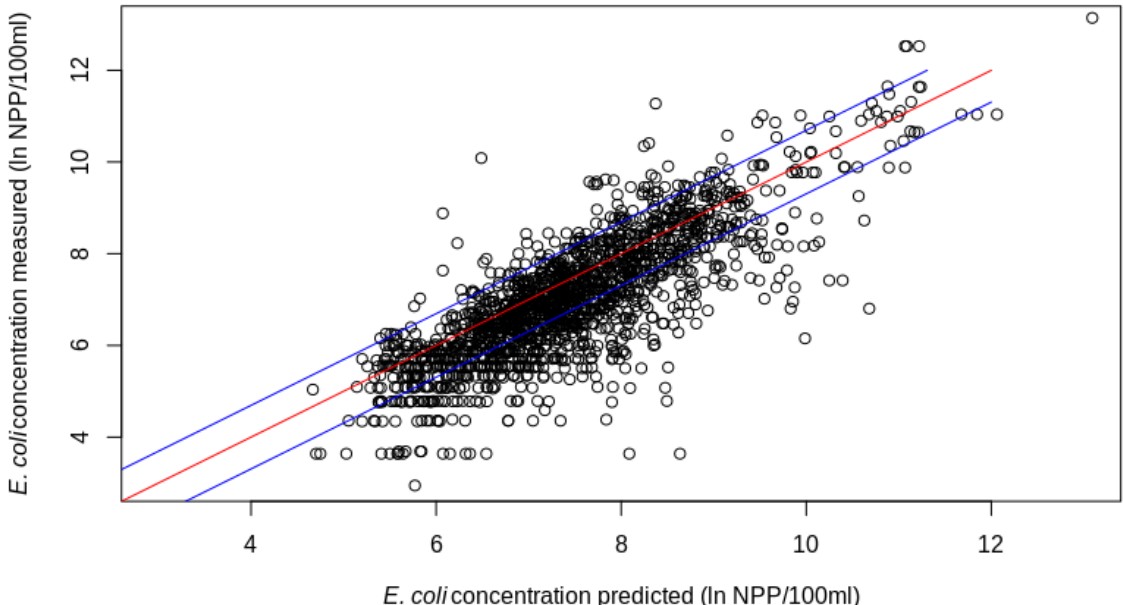

**Figure 5.** Relationship between the *E. coli* concentration predicted by the RF-based model and the measured concentration. The white circles indicate the values. The red line indicates theoretical values corresponding to an accurate prediction of the model compared to the measured values for the ten testing trials. Blue lines indicate the 50% confidence interval.

### 3.4. Identification of the Weaknesses in the Dataset

Different methods can be used to improve the input datasets. Some studies focuses on finding the best combination of input variables to improve the algorithm's predictions (see, e.g., in [15,44]). However, weak features also represent a powerful source of information, that can be used in combination with the features that are adequate for learning the target concept [45]. Thus, in our study, we propose to use the second strategy. For this purpose, the prediction limits and biases of the RF-based model were further examined in order to identify among the physicochemical and hydro-meteorological variables the weaknesses in the training and testing datasets.

We hypothesized that the variability induced by the low representativeness of some parameters can affect the predictive capacity of the model. In order to identify the key parameters allowing a reasonable estimation of *E. coli* concentrations and those leading to an inaccurate estimation, the predicted values were separated in two datasets (inaccurate and reasonable estimations) based on the MAPE indice 50% threshold. Then, an analysis of the relationship between the different physico-chemical and hydro-meteorological variables and the predicted values was carried out on the inaccurate and reasonable datasets. We assumed that a significant difference in the coefficient correlation between the two datasets would point out the variables that had an impact on the outcome of the model but needed optimization. To compare the correlation coefficients obtained with the reasonable and inacurrate datasets, a *t*-test was used ($n = 10$). The p-values obtained are displayed in (Table 1).

**Table 1.** Correlation coefficients (average $r_s$ ± SD) for the relationship between the values of *E. coli* predicted by the RF model (reasonable and inaccurate) and the environmental variables. Significant coefficients are indicated with a * (coefficient significance test $p < 0.05$). Significant differences between the correlation coefficients of the two datasets are indicated as *t*-test *p*-values < 0.01. MAPE values were used to identify reasonable (less than 50%) and inaccurate (over 50%) estimations of *E. coli* concentrations obtained with the RF model during the ten testing trials.

| Parameters | Reasonable Predictions $r_s$ | Inaccurate Predictions $r_s$ | *p*-Value |
|---|---|---|---|
| Water temperature | −0.17 ± 0.05 | −0.28 * ± 0.07 | 0.001 |
| Conductivity | −0.05 ± 0.11 | −0.18 ± 0.09 | 0.009 |
| Turbidity | 0.42 * ± 0.07 | 0.39 * ± 0.08 | 0.43 |
| TSS | 0.43 * ± 0.09 | 0.40 * ± 0.04 | 0.42 |
| $NH_4^+$ | 0.54 * ± 0.06 | 0.48 * ± 0.07 | 0.05 |
| TKN | −0.03 ± 0.08 | 0.001 ± 0.06 | 0.26 |
| Number of dry days | −0.10 ± 0.09 | −0.01 ± 0.09 | 0.02 |
| 24 h cumulative rainfall of the day | 0.09 ± 0.10 | −0.02 ± 0.11 | 0.02 |
| 24 h cumulative rainfall of the previous day | 0.17 ± 0.08 | 0.03 ± 0.10 | 0.002 |
| River flow | 0.54 * ± 0.09 | 0.39 * ± 0.09 | 0.001 |

Turbidity, TSS, and $NH_4^+$ were important predictors (significant $r_s$ above 0.40), and no significant differences in the two datasets occurred (*t*-test, $p \geq 0.05$, Table 1). We classified these parameters as having an impact of the RF-model output, with no urgent need for additional data. The river flow also contributed to the model output (significant $r_s > 0.40$), but there was a significant difference between the two datasets (*t*-test, $p < 0.01$, Table 1). It was thus considered as an important parameter that needs additional data. Finally, the water temperature, the conductivity, the 24 h cumulative rainfall of the previous day (Table 1, *t*-test, $p < 0.01$), as well as for the number of dry days after the last rain and 24 h cumulative rainfall of the day (Table 1, *t*-test, $p < 0.05$) showed weak correlations with *E. coli* values, but a difference between the two datasets. As it is not certain if these weak correlations are an artifact due to the skewed distribution of these parameters or if these

parameters are just bad predictors, we decided to further explore the parameters with a highly significant difference in the correlation obtained with the reasonable and inaccurate estimates. Thus, for the river flow, temperature, the conductivity, and the 24 h cumulative rainfall of the previous day (*t*-test, $p < 0.01$), it was considered that additional data were needed to provide the dataset with enough information to reduce the uncertainty in the model's estimates. The reasons for this uncertainty may be that the measurements have not yet been tested and it is not yet known whether the model will be able to reasonably estimate the *E. coli* concentration, or that the distribution of the data is skewed and the correlation of some environmental variables with the *E. coli* concentration is not yet obvious.

The next step was to identify the value ranges of the four parameters that needed extra measurements to efficiently complete the training and testing datasets. A deeper understanding of the behavior of these parameters in the model should help optimizing the sampling process while minimizing additional cost and efforts of sample collection and analysis. The temperature is the parameter for which the reasonable predicted values of *E. coli* densities covered pretty well the whole range of values (17.6–26.5 °C) (Figure 6). For the conductivity (data range: 430–657 μS/cm) and the flow (data range: 4–101 (m$^3$/s)) the distribution of the reasonable estimates was not regularly disseminated along the data range, and the 24 h cumulative rainfall of the day there was only 4 reasonable values in the (data range: 0–35.4 mm) (Figure 6). The Figure 6, is a valuable tool to identify which data are missing in the data range, and thus help to determine where the sampling efforts should be carried out.

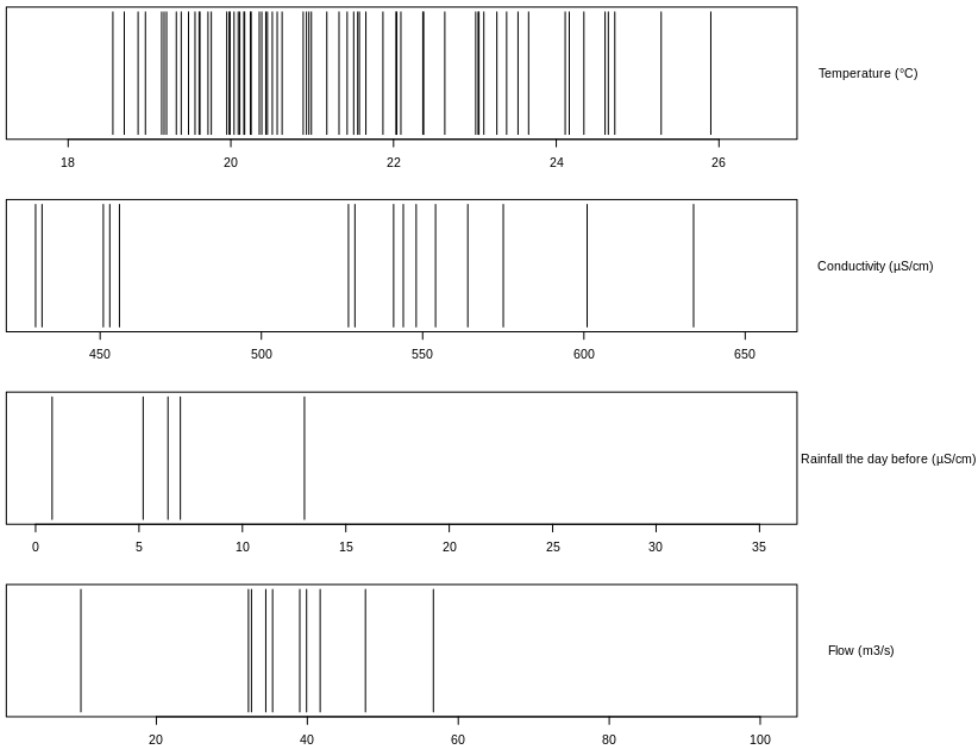

**Figure 6.** Visualization of the values that need enrichment in the dataset for the temperature, conductivity, 24 h cumulative rainfall of the previous day and the river flow. The abscissa displays the value range of each parameter. Predicted values giving a reasonable estimation are visualized with solid black bars, white spaces represent the values that need further enrichment in the dataset.

In our study, the RF-based model produced a versatile modeling in prediction. Based on this observation, we were able to identify a set of parameters and values needed to complete the dataset. An alert system based on the analysis of the reasonable and inaccurate estimates would be a valuable tool for stakeholders to optimize their sampling and measurement efforts. However, manual sampling and laboratory analysis may be too costly

and labor-intensive to realistically implement the training dataset. A network of sensors allowing continuous monitoring of physico-chemical parameters and the monitoring of rainfalls as well as dry weather, could help in optimizing the sampling. Such approach may help developing models able to adapt under environmental perturbations such as accidental pollution or heavy rainfalls (>30 mm), which are usually under-represented in the datasets due to their scarcity, and/or the fact that weekly/monthly routine survey often miss such events.

## 4. Automated Data Collection

From the results, it is clear that the machine learning models are capable of delivering interesting results, as long as one can provide enough good-quality data. Thus, the use of data sensors in addition to manual collection should be investigated as means of feeding these models. Concerning the water quality parameters that we have investigated in this work, there are a myriad of sensors that could perform their collection with acceptable data quality. Those sensors may vary in price, accuracy, usage, and lifespan, among other characteristics, as they were extensively studied in [46,47]. Therefore, to incorporate sensors as a permanent brick in the data collection system, further studies must be conducted to determine their optimal and sub-optimal numbers to be deployed on a given site, the expected accuracy and the available budget for their acquisition. In this direction, Abegaz et al. [46] have thoroughly discussed the nature of different sensors (piezoelectric, optical, etc.) and how they fit for different use cases, while Kruse [47] provides interesting inputs concerning their usages for different use cases.

One strategy for monitoring the bathing water quality and deciding when to open or close a bathing site is to use online measurement systems that detect Beta-D-Galactosidase or Beta-D-Glucuronidase activity. For instance, the ColiMinder automated measurement system (Vienna Water Monitoring, VWM GmbH) [48], ALERT system (Fluidion) [49], Colifast ALARM™ [50], and TECTA-B16 (Endetec, Veolia) [51] have been demonstrated to be useful to monitor *E. coli* in rivers, but the price of these devices may be economically prohibitive for numerous cities, as one unit may cost up to tens of thousand of euros. Alternatively, the use of sensing technologies to measure proxies or surrogate parameters procures high frequency, precise, and accurate data. Based on electrodes, fluorescence, colorimetry, wet analytical chemistry, or flow cytometry techniques, these devices are deployed at fixed strategic locations [52]. However, these sensors are often costly ($\sim$10 K euros unit price), for instance, multiparameter sensors such as Proteus Multi-parameter Water Quality Sensor based on tryptophane-like fluorescent detection or sensor platforms measuring physico-chemical proxies (such as YSI, Sea-Bird or NKE instrument) are often used to monitor water quality.

One interesting way of integrating a network of sensors to data collection is to build an Internet of Things (IoT) network, mixing high-quality (expensive) and medium-quality (cheaper) devices capable of delivering real-time analysis. In comparison, cheaper sensors can be used to deliver good enough approximations of the correct data. For instance, the KnowFlow platform [24], based on Arduino computers and IoT long-range communication, can be a significant addition to the network. A recent review of low-cost sensors is provided by Wang et al. [53].

Concerning the deployment of these heterogeneous sensors, some approaches exist to maximize the quantity and quality of gathered data. The collection system may rely on (i) deterministic deployment, where sensors position is calculated before the collection begins, based on the environmental and economic conditions [54]; (ii) random deployment, in the case where areas are hard to achieve and to position sensors [55]; or (iii) hybrid deployment, a mix of aforementioned approaches, which is used indicated to very large networks, covering vastly heterogeneous areas [56]. Some studies have investigated this topic, with a further analysis on the advantages of IoT networks to enhance data collection [57,58]. For instance, in [57] the authors proposed a methodology to derive the

optimal placement of sensors in an aquatic environment, based on a "divide-and-conquer" approach, which could reduce the complexity of this task for large scenarios.

The deployment of sensors will heavily depend on the battery lifespan of devices, as much as on their communication range. Therefore, IoT-based measurement networks should be based on Low-Power Wide-Area Networks (LPWAN) technologies, as LoRaWAN, Sigfox, or NB-IoT. Such technologies allow communications range up to kilometers and ensure very low energy consumption, when compared to 4G, Wi-Fi, or Bluetooth networks [59]. Users can also consider the utilization of new 5G cellular technology, which is adapted for large-scale sensor networks and IoT communications [60].

One remaining challenge to enhance the use of IoT networks for water quality assessment is the real-data collection and visualization mechanisms. For example, Grafana allows users to analyze sensor metrics through dashboards, messaging and alerts in real time [61]. The Elastic stack application allows a deeper analysis of data logs and provides so-called intelligent dashboards, capable of adapting screens to environmental, economic or user contexts (e.g., what a researcher sees is not what a common user would see) [62]. In [58], the authors developed an IoT-based system within a town, capable of sensing the environmental parameters and effectively delivering real-time information on water quality. This clearly shows that the automation of the collection process is possible and viable for the estimation of water quality in urban sites [63].

Although the use IoT network composed of heterogeneous sensors is an interesting solution to enhance surveillance systems, the use of low-quality devices must be taken with caution due to their less accurate results. Therefore, the calibration of sensors remains an important issue to be investigated. As discussed in [46], the errors, margins, and durability of devices vary a lot. Therefore, an automated data collection must take into account a mechanism to estimate which sensors are no longer in optimal operation conditions, which is more likely to happen to low-quality models. One simple solution consists in compare their output to nearby high-quality devices and analyze when important deviations occur. More complex solutions would consist in estimating their lifespan based on already collected data to perform changes preemptively.

## 5. Conclusions

In this paper, we proposed a framework and statistical indicators to select among a toolbox of six supervised learning algorithms (KNN, SVM, DT, RF, Bagging, and AdaBoost) the most suitable model for the prediction of fecal indicator bacteria in an urban river. This framework was successfully applied to the Marne River (Greater Paris, France). Nevertheless, with regard to the actual dataset, *E. coli* concentration could not be predicted in all contexts (53.2 ± 3.5% of inaccurate predicted values). This result illustrates well the fact that predicting the microbial quality of surface waters in urban rivers remains complex. Refining the models to be able to adapt to environmental changes represents a future challenge in the context of the global change which may increase the frequency of extreme rainfalls and floods [40]. Further amelioration and testing of predictive models is needed to reproduce and predict the temporal and spatial dynamic of fecal indicators in changing and complex aquatic environments. As our dataset was not representative of all the possible values in the data range, some values have not yet been trained or tested by the RF-based model. For these values, it is not clear yet whether our model is able to estimate the *E. coli* concentration in a reasonable way at the moment. To address this problem, we proposed a strategy and tools to help improving the quality and quantity of the training data. The distribution of the accurate values along the data range of each parameter seems an appropriate approach to identify which additional data are needed for which parameter, in order to achieve a good predictive efficiency.

Acquiring additional data is usually costly because it is a manual process that requires human action. As a consequence, our proposed approach aims to optimize the sampling process. It requires focusing on the following points: (i) How and where to use of microbiological high-quality monitoring systems to feed itself. (ii) How to install low cost

physico-chemical sensors on an IoT network for the prediction of microbiological quality. (iii) When to perform sampling by human operators when the model fails to correctly estimate the *E. coli* concentration and the microbiological quality of surface water?

Overall, the proposed framework will help rationalize and optimize the sampling effort, thus saving time and cost of microbiological analyses.

**Author Contributions:** All authors contributed to the manuscript: Conceptualization, S.S., T.W.M.D.A. and F.S.L.; methodology, M.N., S.S. and T.W.M.D.A.; formal analysis, M.N.; resources, A.J.; data curation, M.N., A.J., P.S. and F.S.L.; writing—original draft preparation, M.N.; writing—review and editing, F.S.L., S.S., T.W.M.D.A., A.J. and P.S.; supervision, T. Abreu, S.S. and F.S.L.; project administration, A.J.; funding acquisition, A.J. This manuscript is the final scientific paper of P.S. who suddenly died on August 2021 at the age of 64 years old. During his career, he considerably contributed to the field of aquatic microbial ecology, specially the study of microbial quality of surface waters. This paper is thus specially dedicated to Pierre Servais, who diligently and meticulously contributed to improve the manuscript while he was courageously fighting a cancer. All authors have read and agreed to the published version of the manuscript.

**Funding:** Data were acquired through the annual budget of the Syndicat Marne Vive. Manel Naloufi phD grant is provided by the City of Paris and the French Association Nationale de la Recherche et de la Technologie.

**Data Availability Statement:** Datasets are deposited in the CapGeo database of a working group directed by the City of Paris to study the water quality of the Seine and the Marne river. This dataset is not yet openly accessible.

**Acknowledgments:** We thank the Departmental Councils of Val-de-Marne and of Seine-Saint-Denis (France) and the city of Paris for their contribution to the dataset. We are grateful to Lamine Amour for his constructive preliminary work. We are grateful to Miguel Gillon-Ritz for his sound advices and for the access to CapGeo database (City of Paris, Direction de la Propreté et de l'Eau—Service Technique de l'Eau et de l'Assainissement). We thank Miguel Gillon-Ritz and Marion Delarbre for their kind welcome and supervision at the Service Technique de l'Eau et de l'Assainissement (City of Paris).

**Conflicts of Interest:** The authors declare no conflicts of interest.

## Appendix A

**Table A1.** Descriptive statistics of the parameters.

| Parameters | Mean | Standard Deviation | Minimum | Maximum |
|---|---|---|---|---|
| Water temperature | 21.77 | 1.59 | 17.60 | 26.50 |
| Conductivity | 537.36 | 56.09 | 430.00 | 657.00 |
| Turbidity | 7.91 | 7.33 | 0.12 | 132.00 |
| TSS | 9.76 | 9.13 | 0.90 | 190.00 |
| $NH_4^+$ | 0.12 | 0.07 | 0.03 | 1.11 |
| TKN | 0.72 | 1.08 | 0.15 | 33.70 |
| Number of dry days | 4.80 | 5.72 | 0.00 | 27.00 |
| 24 h cumulative rainfall of the day | 0.97 | 2.90 | 0.00 | 26.00 |
| 24 h cumulative rainfall of the previous day | 1.86 | 4.69 | 0.00 | 35.40 |
| River flow | 41.68 | 10.59 | 4.00 | 101.00 |

**Table A2.** Average and standard deviation of the statistic metrics (RMSE, MAE, RDP) obtained with each model during the ten testing trials.

| Model | KNN | RF | DT | SVM | AdaBoost | Bagging |
|---|---|---|---|---|---|---|
| RMSE | 0.41 ± 0.28 | 0.37 ± 0.20 | 0.54 ± 0.29 | 0.53 ± 0.48 | 0.53 ± 0.28 | 0.38 ± 0.19 |
| MAE | 0.09 ± 0.03 | 0.09 ± 0.02 | 0.14 ± 0.05 | 0.13 ± 0.05 | 0.10 ± 0.03 | 0.14 ± 0.06 |
| RDP | 1.60 ± 0.49 | 1.91 ± 1.65 | 1.12 ± 0.36 | 1.32 ± 0.22 | 1.28 ± 0.62 | 1.77 ± 1.62 |

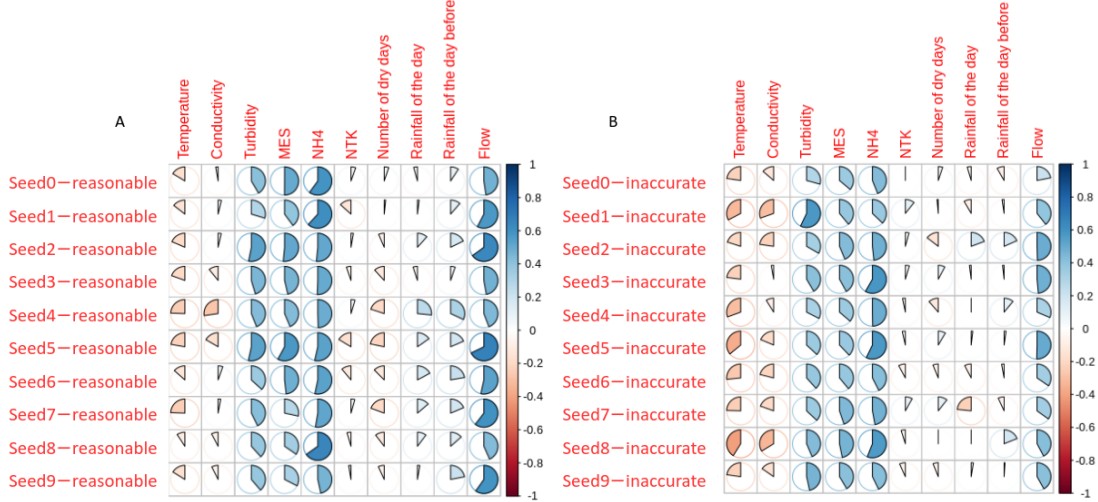

**Figure A1.** Correlation analysis between water quality parameters and *E. coli* concentration estimated by RF for (**A**) reasonable estimation and (**B**) inaccurate estimation of *E. coli* (*n* = 10).

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
