# Peer review of "Evaluating the Performance of Machine Learning Approaches to Predict the Microbial Quality of Surface Waters and to Optimize the Sampling Effort"

_water, doi:10.3390/w13182457_

Round 1

Reviewer 1 Report

This paper discusses a machine-learning-based model to predict an estimate of water quality in the Marne River (Greater Paris, France). Six supervised learning algorithms (KNN, SVM, DT, RF, Bagging and AdaBoost) are tested to verify their ability to estimate the E. coli concentration. Three statistical metrics are used to evaluate the performance of these prediction models. However, as the authors point “it is not clear yet whether our model is able to estimate the E. coli concentration in a reasonable way at the moment.” (Lines 399-400). What is the value of similar studies after catastrophic flood this summer in Europe?

The paper is very difficult for understanding and demands numerous text and figure explanations.

  1. In formulae (1), (2), (4), the replacements of the notations of yi by yi , and yi by yi’ are necessary.
  2. What is the formula for the standard deviation of the tested dataset (SD)?
  3. (Lines 189-190) “For all statistical tests, the significance level was based on 5% and 1%.”

What are certain statistical tests with the significance level based on 5% and what are based on 1%?     

  1. Figure 3 - the notations and plots are not clear. What parameter is located at the abscissa-axis? What are the black and grey areas at the figures? Horizontal lines? Black and white circulars?
  2. Figure 4 - the notations and plots are not clear. What parameter is located at the ordinate-axis? See also similar questions to Fig. 3.
  3. Figure 5 - How do you define reasonable and inaccurate estimations? What are the estimations?
  4. Table 1 - How do you define reasonable and inaccurate predictions? What is the p-value?
  5. (Line 300) What is the t-test?
  6. Figure 7 - the notations and plots are not clear. What do vertical black and grey lines define?
  7. Incomplete references Nos. 1, 13, 19,20, 28, 32.

Reviewer 2 Report

  1. In the methodology part, the unit for the concentration of E.coli bacteria - NPP is given. In the results - MPN. Please standardize
    2. How many neighbours with the KNN method were taken into account (1,3 or 5?)
    3.In conclusion,  please add the difficulty in predicting the microbiological quality of water
  2. 4.Have the authors thought about modifying the methodology and compacting the sampling sites? Wouldn't such a procedure improve the forecasting results? In the presented material, the distances between sampling attempts are quite large. Please add it to the discussion.

Round 2

Reviewer 1 Report

The authors have performed very significant work on improvement of the original paper.

Nevertheless, the next moments should be corrected:

  1. (Line 202) It must be corrected to (MAPE ≥ 50%).
  2. (Line 207-214) It should be more clearly pointed out what is the t-test with p-value is used, because there are several types of the t-test.
